# Adsorption of Methyl Orange from Water Using Chitosan Bead-like Materials

**DOI:** 10.3390/molecules28186561

**Published:** 2023-09-11

**Authors:** Haya Alyasi, Hamish Mackey, Gordon McKay

**Affiliations:** Division of Sustainable Development, College of Science and Engineering, Hamad Bin Khalifa University, Education City, Qatar Foundation, Doha 24144, Qatar

**Keywords:** chitosan beads, methyl orange adsorption, kinetic modelling, equilibrium studies, regeneration

## Abstract

Natural product waste treatment and the removal of harmful dyes from water by adsorption are two of the crucial environmental issues at present. Traditional adsorbents are often not capable in removing detrimental dyes from wastewater due to their hydrophilic nature and because they form strong bonds with water molecules, and therefore they remain in the dissolved state in water. Consequently, new and effective sorbents are required to reduce the cost of wastewater treatment as well as to mitigate the health problems caused by water pollution contaminants. In this study, the adsorption behaviour of methyl orange, MO, dye on chitosan bead-like materials was investigated as a function of shaking time, contact time, adsorbent dosage, initial MO concentration, temperature and solution pH. The structural and chemical properties of chitosan bead-like materials were studied using several techniques including SEM, BET, XRD and FTIR. The adsorption process of methyl orange by chitosan bead materials was well described by the Langmuir isotherm model for the uptake capacity and followed by the pseudo-second-order kinetic model to describe the rate processes. Under the optimal conditions, the maximum removal rate (98.9%) and adsorption capacity (12.46 mg/g) of chitosan bead-like materials were higher than those of other previous reports; their removal rate for methyl orange was still up to 87.2% after three regenerative cycles. Hence, this chitosan bead-like materials are very promising materials for wastewater treatment.

## 1. Introduction

Dyes, such as methyl orange, are major pollutants discharged by the textile, pharmaceutical, food and printing industries, causing water pollution and difficulty in processing the wastewater [1,2]. These dyes are toxic in nature even at very low concentrations and can even cause cancer in humans [3]. Methyl orange has low biodegradability and is stable in aqueous solutions, which is the reason why it is difficult to remove methyl orange from wastewater [4]. Traditional water treatment methods such as coagulation, filtration, oxidation, sedimentation, biological digestion and adsorption using activated charcoal are ineffective in removing methyl orange [5,6]. Clays such as montmorillonite, smectite clays, kaolinite and illite are used for the removal of methyl orange from wastewater [7]. Chitosan has been used widely to remove methyl orange from wastewater. Chitosan contains a primary amine group, which facilitates strong electrostatic interaction with cationic dyes such as methyl orang [8]. The extent of removal of the dye depends on the surface charge of the adsorbent [9]. Furthermore, chitosan has been used extensively in flake, powder and composite forms to successfully remove several classes of contaminants, including, acid dyes [10], basic dyes [11], arsenic [12], emerging contaminants [13,14], endocrine disrupting compounds [15], humic acid [16], a wide range of heavy metals [17,18] such as cadmium [19] and lead [20], mercury [21] and molybdate [22].

Several different forms of chitosan-based beads, flakes, gels, sponges and nanofibers have been prepared and studied for identifying the adsorption capacity of persistent pollutants such as methyl orange [23,24]. However, the practical application of gels, flakes and nanofibers is questionable for larger-scale column adsorption systems.

To our knowledge, few studies on the adsorption of organic pollutants by nonmodified chitosan bead-like materials have been reported. Moreover, few works have been complemented by experiments on pollutant desorption. However, it is important to prove the reusability of these materials if they have to be integrated into a water treatment process. The objective of this paper is to study the chemical and physical adsorptive features of the chitosan bead-like materials and to identify the main parameter controlling the fixation of methyl orange molecules by chitosan bead-like materials. The approach will be implemented through two types of experiments: isotherm and kinetics using the batch method. A characterization study of chitosan bead-like material using SEM, FTIR and XRD before and after adsorption was carried out. Finally, the assessment of the regeneration performance of chitosan over at least five cycles has been performed to provide an efficient and economical process of water purification.

## 2. Results and Discussion

### 2.1. Characterization of the Adsorbent Material

#### 2.1.1. Physical Properties

From Table 1, the surface area of chitosan bead-like adsorbents decreased from 0.762 m^2^ g^−1^ to 0.296 m^2^ g^−1^ with the increasing particle size from chitosan powder (335–500 µm) to chitosan beads (1.00 mm). The total pore volume of the chitosan bead-like adsorbents also decreased from 2.96 × 10^−2^ cc g^−1^ to 1.86 × 10^−2^ cc g^−1^ with the increasing particle size. This indicates that there is a small increase in the crystallinity of the chitosan powder adsorbents, as well as an increase in the free RNH_2_ sites in the small gaps of the chitosan powder, which indicates a larger pore volume, according to certain studies [25]. Due to this larger pore volume, powder chitosan forms have high crystallinity compared to bead-like chitosan forms.

#### 2.1.2. SEM

Chitosan bead-like materials and MO–chitosan structures were observed by scanning electron microscope. The SEM images for all the samples were found to be different. It was found that MO had several different structures from micron to nanoscale, while the chitosan exhibited a slightly wrinkled network structure. A nano-needle structure was visible in MO–chitosan-based material, which indicates that methyl orange is adsorbed on the chitosan beads (Figure 1). The colour intensity of the dyes in Samples 1 and 2 also differs, which means that methyl orange is adsorbed on the chitosan beads. Similar results were reported in other study [26].

#### 2.1.3. FTIR

The FTIR spectrum, shows a prominent band for chitosan-based materials before and after adsorption occurs in the 3200–3600 cm^−1^ region corresponding to -OH and -NH_2_ stretching. IR bands near 2900 cm^−1^ correspond to aliphatic C-H stretching, and the signature band at 1635 cm^−1^ corresponds to -NH_2_, while the band at 1380 cm^−1^ corresponds to -CH symmetric bending vibration of -CHOH. The IR band near 1100 cm^−1^ was attributed to C-O stretching of C-O-H groups. Since 20–25% of chitosan contains acetyl groups, the corresponding carbonyl signature above 1660 cm^−1^ may be assigned to acetyl group. The main differences can be observed in the range 1000–1600 cm^−1^, where the width of the broadband is more reduced, and the broadband has sharper and more decided edge (Figure 2). The imposed IR spectra of chitosan and chitosan with methyl orange clearly suggest a structural change in the amino groups present in chitosan. This indicates that the amino groups bind with sulphonate groups of MO dye. The same trend of results has been reported in other studies [8,27].

#### 2.1.4. XRD

The structural properties of chitosan were investigated by XRD before and after adsorption. The XRD patterns provide information on the degree of crystallinity of the sample. The XRD pattern of chitosan beads displayed a broad peak with low intensities of high-order diffraction peaks (Figure 3a). This suggests a poor chitosan chain crystallinity, while the XRD peaks of chitosan and chitosan with methyl orange dye do not differ much [28].

In the XPS analysis, the samples of chitosan and chitosan with methyl orange reflect the presence of C, N and O, but in differing ratios. The ratio of C:N:O in Sample 1 is 14:1:5, while it is 7.5:1:3.8 in Sample 2. Sample 3 shows the presence of sulphur and sodium; however, these signals are not present in Sample 2 (Figure 3b). The changes in the concentration of carbon in the first and second samples are mainly because of the –C–C bonds, while the –N–C=O and –C–N bonds are constant.

### 2.2. Batch Adsorption Experiments

#### 2.2.1. Effect of Initial Concentration

The effect of various initial concentrations of MO solution on adsorption capacity are illustrated in Figure 4. Increasing the MO concentration decreased the removal percentage but increased the specific adsorption capacity. The maximum removal percentage of MO dye was 98.8%, and it was achieved with a 20 mg/L MO concentration. At low concentrations, the ratio of dye molecules to the accessible active site is low; therefore, more active sites were available for MO molecules to access. Consequently, this increases the removal percentage of MO. In contrast, fewer active sites are available for MO molecules at high MO concentrations due to the ratio of high dye molecules to the surface active sites leading to a low removal percentage but a higher dye mass adsorption capacity [26].

Although the removal percentage decreased with increasing the initial concentration, the quantitative amounts of dye adsorbed increased. With increased concentration, MO molecules electrostatically repel each other, leading to a competition for the active site on chitosan. A similar effect was observed in different dye adsorption studies [29].

#### 2.2.2. Effect of Contact Time

Figure 4 depicts the relationship between initial MO concentrations and equilibrium time. For initial MO concentrations of 20, 60, and 100 mg/L, the times required to reach equilibrium were 20, 25, and 30 min, respectively. As MO concentrations increased, the MO molecules covered/blocked the adsorbent surface, increasing the surface hindrance for adsorption with time. The MO adsorption onto chitosan was very intense and reached equilibrium quickly at very low concentrations [8].

#### 2.2.3. Effect of Adsorbent Mass

By varying the dose of chitosan bead-like materials from 0.01 to 0.7 g, the effect of the adsorbent mass on methyl orange adsorption was studied. The percentage removal of MO and the uptake capacity of chitosan are represented in circle and triangle points, respectively, in Figure 5. The removal percentage of MO increased with the increase in the adsorbent’s amount. The maximum percentage of MO removal was 98.8, and it was achieved using 0.01 g of chitosan. There are more available active sites and more functional groups for the dye molecules to adsorb due to the greater surface area of the chitosan adsorbent. In contrast, the uptake capacity (mg of MO adsorbed/g of chitosan) decreased with the increase in the adsorbent’s amount. Inaccessible or aggregated active sites reduce the availability of active sites for MO, thus decreasing the uptake capacity.

#### 2.2.4. Effect of Agitation

The effect of the stirring rate on MO adsorption is shown in Figure 6. The effect of agitation speed on methyl orange adsorption was studied by changing the stirring rate from 0 to 300 rpm for 120 min. The results depict that the uptake capacity of MO by chitosan increased as the stirring rate increased. The maximum adsorption capacity was 14.24 mg of MO/g chitosan, which was achieved at 200 rpm (Figure 6). According to Ruthven (1984), increasing the stirring rate causes a reduction of the liquid film thickness around the bead and mass transfer resistance [30]. Therefore, it facilitates dye diffusion and increases the adsorption capacity. The adsorption capacity of the adsorbents does not further increase beyond this, indicating that film thickness has no significant effect when the agitation rate is greater than 200 rpm [31]. The decreased value in capacity at 300 rpm is more challenging to explain. However, it may be caused by the increased turbulence in the solution around the chitosan particle, keeping the MO molecules in solution and restricting the formation of possible surface-complexation bonding between MO and chitosan.

#### 2.2.5. Effect of the Solution pH on MO Dye Sorption

The effect of pH on the adsorption process has been investigated in this study at various pH values from 2 to 10. The pH of MO solution was adjusted using 0.1 M of HCL and NaOH. The pH is a significant parameter influencing the whole adsorption process, it affects the dissociation of functional groups on the adsorbent active sites, the surface charge of the adsorbent and the chemistry of MO solution. The highest removal capacity (98.8%) was achieved at a pH of 5. This is attributed to the fact that surface charge is positive at lower pH values (pH < 2), which makes H^+^ ions effectively compete for dissociated sulfonate groups of MO dye, causing a decrease in adsorption. However, between pH of 3 and 6, the number of hydrogen ions decreased as the pH increased from 2 to 6. Therefore, the adsorption process was facilitated due to the electrostatic interaction between negatively charged anionic dye and positively charged chitosan amino groups, –NH_3_^+^. When pH > 6, the adsorption process slows down due to the competition of OH– ion now present in the basic solution of the adsorption sites with anionic dye ions. At a pH of 8 and 10, a saturation point is reached (Figure 7). This may be attributed to the stability of the amino group present in the chitosan beads at higher pH, resulting in poor interaction with the sulphonic acid group present in the methyl orange. The optimum pH in this study was around pH 4–6. Similar results were observed in acid dye adsorption onto cross-linked chitosan [6].

#### 2.2.6. Effect of Temperature

A plot of removal percentage of MO at different adsorption temperatures is shown in Figure 8. The results reveal that the removal percentage toward MO increased gradually at first, taking 10 min to reach the equilibrium for all temperatures examined. From the results, it can be seen that, above 30 °C, the higher temperatures have only a small effect. It has been reported that the variation in the adsorption temperature of dye removal does not have a significant effect on the overall decolourization process [32].

### 2.3. Adsorption Isotherms

The purpose of the current research is to compare and contrast different equilibrium systems for optimal MO adsorption on chitosan-based material by employing the different isotherm models described in the sections above. Eventually, the best fit is described after an analysis of related equations and conditions. The following results are available from a detailed analysis of equilibrium isotherms. The graphs below describe the equilibrium representations of MO adsorption onto chitosan-based material using Langmuir isotherm, Freundlich isotherm, Redlich–Peterson isotherm and Sips isotherm (Figure 9). The isotherm equations and assumptions have been reported previously [33].

The equilibrium data enable the evaluation of chitosan-based material’s structural properties as they describe how the adsorbate molecules interact with the adsorbent, thus providing a more comprehensive understanding about the nature of the adsorption interactions. The Langmuir and Redlich–Peterson isotherms show excellent correlation and both provide an ideal fit (R^2^ = 0.9999) with the experimental data. The correlation coefficient is close to unity in both cases. For the Redlich–Peterson isotherm, the exponent is 1.06, which is close to unity, and when the exponent equals 1.0, the Redlich–Peterson equation reduces to the Langmuir equation. The maximum uptake capacity was 14.29 and 18.12 mg MO/g chitosan for the Langmuir and Redlich–Peterson isotherms, respectively. The single-stage Freundlich equilibrium model is particularly an outlier in this regard, confirming that the mechanism may involve very little heterogeneous adsorption, and may be deficient in terms of its thermodynamic characteristics, for example, little effect of temperature variation on adsorption capacity [34]. The Sips isotherm, which is a combined expression for both Langmuir and Freundlich models, also shows a reasonable correlation with the experimental data. At high adsorbate concentrations, the Sips equation approximates to the Langmuir isotherm model. Although, the Temkin isotherm shows a better correlation than the Freundlich correlation, it will not be considered in this study due to the fact that the model ignores extreme values of concentrations on the low and high ends.

The R^2^ statistic depicts the success of a particular fit when considering variation with respect to data. R^2^ values lie between 0 and 1, and a specific value indicates the proportion of variance explained by the specific model as a percentage. In Table 2, the R^2^ value of the Langmuir and Redlich–Peterson isotherms explains essentially 100% of the variation in data. The Sips isotherm model also depicts a similar case. The Temkin isotherm models follow closely, still existing in the 89–90% percentile. However, the Freundlich single-stage isotherm model values show the least correlation between the model values and predicted values.

As the assessment of non-linear models can be influenced by the statistical measure and nature of the residuals, a number of other statistical fitting equations were used; these included the SSE, HYBRID, MPSD and ARE. The values of all five error functions are presented in Table 2. By comparing the results of the values for the error function, excluding R^2^ result, it is clear that the Langmuir equation is the best representation of MO adsorption on chitosan-based material. The maximum monolayer adsorption capacity was 14.29 mg/g. Figure 8 shows the general effect of temperature on MO adsorption on chitosan, and it was concluded that the adsorption process is not significantly dependent on the solution temperature. These results suggest that MO–chitosan interaction must be an endothermic process.

The isotherm results will be further used in the discussion regarding the mechanism of MO adsorption onto chitosan-based materials.

### 2.4. Kinetic Studies at Variable Initial Concentrations

The effect of contact time has been studied by the application of several kinetic models to the experimental contact time results. The model equations and their assumptions have been reported in previous studies [35,36]. By comparing the results tabulated in Table 3, it can be seen that the pseudo-second-order model fits the experimental data; the SSE, HYBRID, MDSP and ARE values are relatively small, which indicates that the models explain more than 80% of variability of the adsorption process. These results also indicate that the pseudo-second-order model represents a better fit with the experimental data for MO adsorption on chitosan.

In terms of the adsorption mechanism, there is enough evidence to state that adsorption does not take place on localized sites, and the interaction between the adsorbent active group ion or site and the adsorbate ion is most likely to take place. The fit with the pseudo-second-order model indicates that the adsorbent surface and MO ions interact, and thus chemisorption is most probably the prevalent adsorption mechanism.

The error function values for the intraparticle diffusion models are the highest in the batch studies. This observation was expected because the intraparticle model was developed for surface and pore diffusion in the internal pores of an adsorbent and, therefore, is dependent on the internal surface area of the pores. The poor fit with the intraparticle diffusion model indicates that it must be the external part of the amino-adsorption sites of chitosan that is also mostly available for adsorption. Most sites are surface sites, so they are available to the MO adsorbate. This is a promising result because adsorption is not limited by intraparticle diffusion, and therefore almost all the sites are available for adsorption. This observation is also supported by the low BET surface area of the chitosan, 0.4 m^2^/g, which indicates a very low porosity for intraparticle diffusion to be effective in the rate controlling step.

### 2.5. Proposed Mechanism

The pH plays a key role in predicating the type of interaction between the adsorbent and adsorbent. Figure 7 is an illustration of MO adsorption on chitosan beads at optimum pH. The optimum pH was around 3–6. At low pH, MO in aqueous solution is first dissolved, and the sulphonate groups of MO (-SO_3_Na) are dissociated and converted to an anionic dye ion and a cationic sodium ion:(1)$${DSO}_{3}Na\overset{H_{2}O}{\to }{DSO}_{3}^{-}+{Na}^{+}$$

Due to the presence of H^+^ at the low and moderate pH levels, the amino groups of chitosan (R-NH2) are protonated:(2)$${RNH}_{2}+H_{2}↔{RNH}_{3}^{+}$$

The adsorption process then proceeds due to the electrostatic attraction between the counter ions (Figure 10):(3)$${RNH}_{3}^{+}+{DSO}_{3}^{-}↔{RNH}_{3}^{+}O_{3}SD$$

### 2.6. Regeneration Study

Regeneration of the adsorbent is an important factor for its sustainable, low-cost use in industrial applications. To assess the regeneration and reusability of this adsorbent, it was subjected to multiple cycles of methyl orange adsorption and desorption. The results are shown in Figure 11. After three cycles of regeneration, the removal efficiency was 87.2%. The small decrease in the adsorption capacity is due to the embedded MO molecules on the chitosan surface. However, the adsorption capacity of chitosan beads was compared with other previously reported adsorbents, and the results are summarized in Table 4.

## 3. Experimental Procedures

### 3.1. Materials and Preliminary Characteristics of Adsorbents

Chitosan powder used was supplied by an Indian Chemical Company (Ahmedabad, India). The sample was used without further purification. It was characterized by an average deacetylation rate close to 85%, and the average molecular weight (MW) was about 50,000–190,000 g mol^−1^. Analytical-grade methyl orange (C_14_H_14_N_3_NaO_3_S; molecular weight 327.33 g/mol) was obtained from Sigma-Aldrich and was used without further purification. Acetic acid and sodium hydroxide were obtained from Sigma-Aldrich (St. Louis, MI, USA). Deionized water was used for all dilutions and solution preparation purposes throughout this study.

### 3.2. Synthesis of Chitosan Bead-like Materials

All 3% *w*/*w* chitosan powder samples (particle size of 335–550 µm) were dissolved in 2% acetic acid under magnetic stirring at room temperature for 6 h. Later, the mixtures were maintained on an orbital shaker overnight to ensure complete dissolution. These chitosan solutions were then added to 2 M NaOH using a burette. The use of the burette helped to control the release of chitosan solution into NaOH, thereby effectively controlling the size of the chitosan beads. This varies from other methods used for chitosan beads preparation, where the size of the chitosan beads was controlled by making a hole in a centrifuge tube and adding the chitosan solution through the tube into NaOH [39]. The obtained beads were then washed to neutral pH using deionized water and acetone before being dried at 60 ± 2 °C in a vacuum oven for 6 h. The samples were kept in a desiccator for future use. The experimental setup for chitosan is demonstrated in Figure 12.

### 3.3. Material Characterization

Various structural and optical characterization methods were studied before and after adsorption using SEM, BET, FTIR and XRD.

### 3.4. Adsorption Isotherm

A total of 0.07 g of chitosan-based material was mixed with 100 mL of various methylene orange (MO) concentrations (1, 2.5, 5, 7.5, 10, 12.5, 15, 17.5, 20, 22.5, 25, 30, 40, 50, 60 and 100 mg/L) in a 125 mL flask. The pH was adjusted to the range 5–6 by using 0.1 mol/L HCl or 0.1 mol/L NaOH solutions. The samples were equilibrated at 30 °C on an orbital shaker for 120 min and 200 rpm. The samples were filtered after adsorption, and a standard calibration curve was used to determine the concentration of the MO in the supernatant liquid. The concentration of MO was taken before and after adsorption in duplicate. The uptake capacity was determined using Equation (4).
(4)$$q_{e}=v(C_{0}-C_{e})/m$$

### 3.5. Batch Adsorption Experiments

#### 3.5.1. Effect of MO Initial Concentration

A fixed mass (0.07 g) of chitosan-based material was mixed with a fixed amount of 100 mL methylene orange at 30 °C. The pH was adjusted to the range 5–6 by using 0.1 mol/L HCl or 0.1 mol/L NaOH solutions. The samples were placed on a stirrer at a variable speed. Sampling was performed by sequential removal of 2 mL aliquots over a time interval of 0–60 min using a range of initial MO concentrations (20, 40, 50, 60 and 100 mg/L). The samples were filtered after adsorption, and a standard calibration curve was used to determine the concentration of the MO in the supernatant liquid. The concentration of MO was taken before and after adsorption and in duplicate.

#### 3.5.2. Effect of Adsorbents Dosage

The effect of adsorbent mass on the adsorption of methylene orange has been studied using the agitated batch sample jar method. Variable masses (0.07, 0.1, 0.2, 0.5 and 0.7 g) of chitosan-based material were added to a fixed amount of 100 mL methylene orange at 30 °C. The pH was adjusted to the range 5–6 by using 0.1 mol/L HCl or 0.1 mol/L NaOH solutions. The samples were placed in the jar, and agitation was achieved using a variable speed stirrer set at a fixed speed of 200 rpm. Sampling was performed by sequential removal of 2 mL samples over a time interval of 0–60 min using 20 mg/L methylene orange solution. The samples were filtered after adsorption, and a standard calibration curve was used to determine the concentration of the MO in supernatant liquid. The concentration of MO was taken before and after adsorption in duplicate.

#### 3.5.3. Effect of Agitation Speed

A fixed mass (0.07 g) of chitosan-based material was mixed with fixed amount of 100 mL methylene orange at 30 °C. The pH was adjusted to the range 5–6 by using 0.1 mol/L HCl or 0.1 mol/L NaOH solutions. The four samples were placed in sample jars and agitated with a variable-speed stirrer to assess the effect of agitation speed—0, 100, 200 and 300 rpm. Sampling was performed by sequential removal of 2 mL samples over a time interval of 0–60 min using 20 mg/L methylene orange solution. The samples were filtered after adsorption, and a standard calibration curve was used to determine the concentration of the MO in supernatant liquid. The concentration of MO was taken before and after adsorption in duplicate.

#### 3.5.4. Effect of MO Solution pH

A fixed mass (0.07 g) of chitosan-based material was mixed with fixed amount of 100 mL methylene orange at 30 °C. The pH was adjusted to different values (4, 6, 8, 10) by using 0.1 mol/L HCl or 0.1 mol/L NaOH solutions. The samples were placed on a stirrer with variable speed. Sampling was performed by sequential removal of 2 mL over a time interval of 0–60 min using 20 mg/L MO solution. The samples were filtered after adsorption, and a standard curve is used to determine the concentration of the MO in the supernatant liquid. The concentration of MO was taken before and after adsorption and in duplicate.

#### 3.5.5. Effect of Temperature

A fixed mass (0.07 g) of chitosan-based material was mixed with a fixed amount of 100 mL of methylene orange at variable temperatures (30, 40 and 45 °C). The pH was adjusted to the range 5–6 by using 0.1 mol/L HCl or 0.1 mol/L NaOH solutions. The four samples were placed on a stirrer with a speed of 200 rpm. Sampling was performed by sequential removal of 2 mL over a time interval of 0–60 min using 20 mL/g methylene orange solution. The samples were filtered after adsorption, and a standard curve is used to determine the concentration of the MO in supernatant liquid. The concentration of MO was taken before and after adsorption in duplicate.

## 4. Conclusions

This study provides a comprehensive understanding of chitosan structural and adsorptive behaviours. Chitosan bead-like materials were prepared and tested for physicochemical properties and removal efficiency towards MO. The results revealed that, for MO adsorption, the best isotherm model was Langmuir. Therefore, MO adsorption on chitosan is a homogenous sorption mechanism, where the sorption of each sorbate molecule onto the surface has equal activation energy. The maximum monolayer capacity was 14.29 mg/g. For the kinetics studies, the pseudo-second-order kinetic model agreed very well with the dynamic behaviour for the adsorption of MO onto chitosan under several different pHs, initial dye concentrations, temperatures, contact times and different adsorbent dosages in the whole range studied. Moreover, intraparticle diffusion is not the rate-determining step, and it could be proved by the deviation from the origin. In addition, a very small specific surface area of chitosan indicates a very low internal porosity. On the contrary, the adsorption process of MO on chitosan is controlled by chemical reactions. For adsorption/desorption cycles using NaOH as a desorption agent, the chitosan bead adsorbent showed good reusability, providing an efficient and economical method for water purification and recovery of value-added materials.

## Figures and Tables

**Figure 1 molecules-28-06561-f001:**
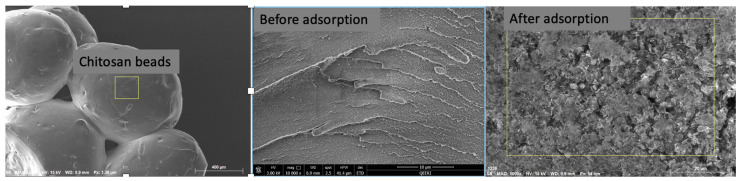
Structural properties of chitosan bead materials before and after adsorption.

**Figure 2 molecules-28-06561-f002:**
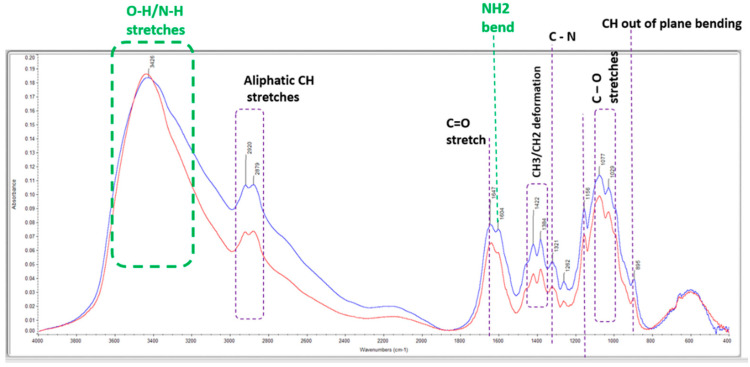
Overlay of FTIR of chitosan structure before adsorption (blue) and MO–chitosan structure after adsorption (red).

**Figure 3 molecules-28-06561-f003:**
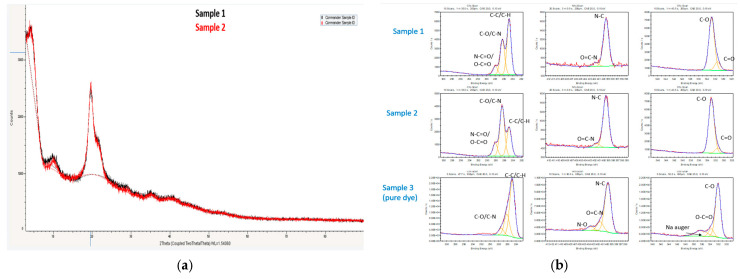
(**a**) Overlay of XRD of Sample 1 and Sample 2 (**b**) XPS analysis of all the three samples.

**Figure 4 molecules-28-06561-f004:**
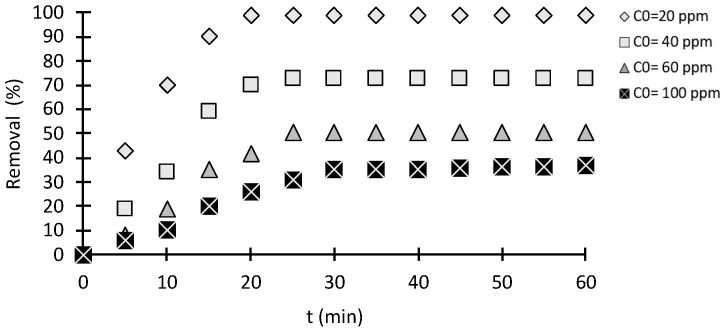
Effect of initial MO concentrations on the adsorption capacity (adsorbent weight: 0.01 g; volume: 100 mL).

**Figure 5 molecules-28-06561-f005:**
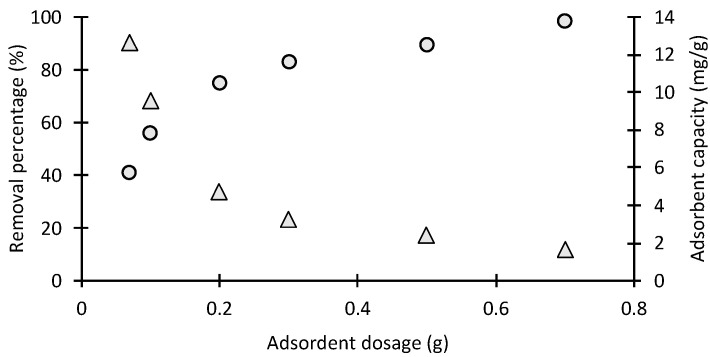
Effect of adsorbent dosage on MO adsorption on chitosan (volume: 100 mL; C_0_ = 20 mg/L) where is triangle and circle dots represent the adsorption capacity (mg/g) and removl efficiency respectively.

**Figure 6 molecules-28-06561-f006:**
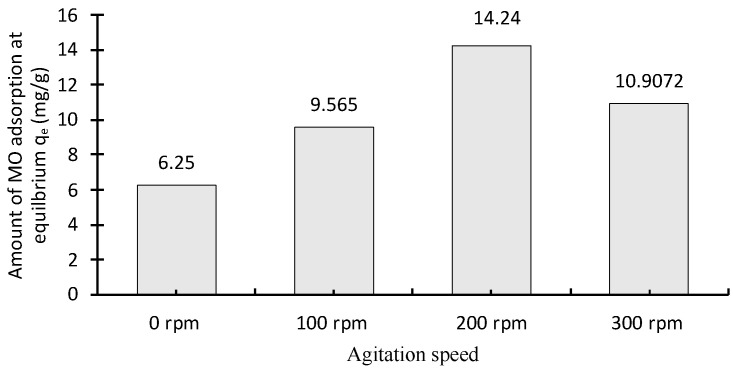
Effect of agitation speed on MO adsorption on chitosan (volume: 100 mL; C_0_ = 20 mg/L).

**Figure 7 molecules-28-06561-f007:**
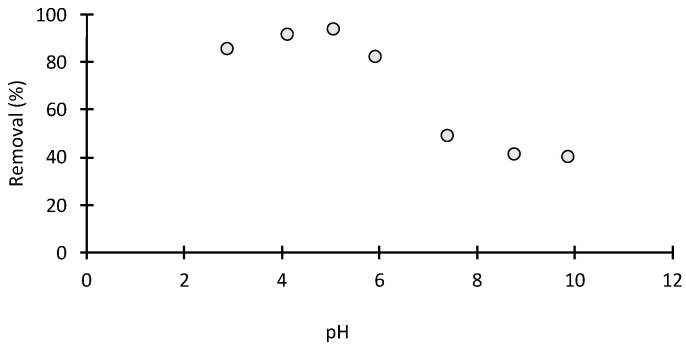
Effect of pH on MO adsorption on chitosan (adsorbent weight: 0.100 g; volume: 100 mL; C_0_ = 20 mg/L).

**Figure 8 molecules-28-06561-f008:**
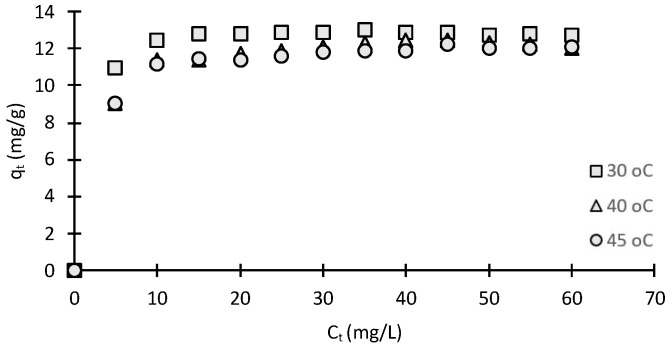
Kinetics of MO adsorption on chitosan at varying temperatures (adsorbent weight: 0.100 g; volume: 100 mL; C_0_ = 20 mg/L).

**Figure 9 molecules-28-06561-f009:**
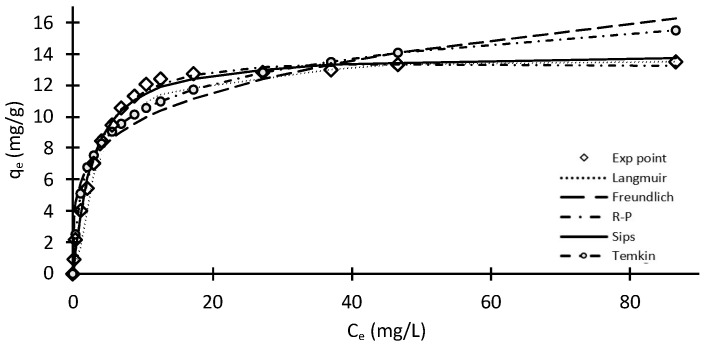
Plot of several isotherm models of the adsorption of MO on a chitosan-based material at pH= 5–6.

**Figure 10 molecules-28-06561-f010:**
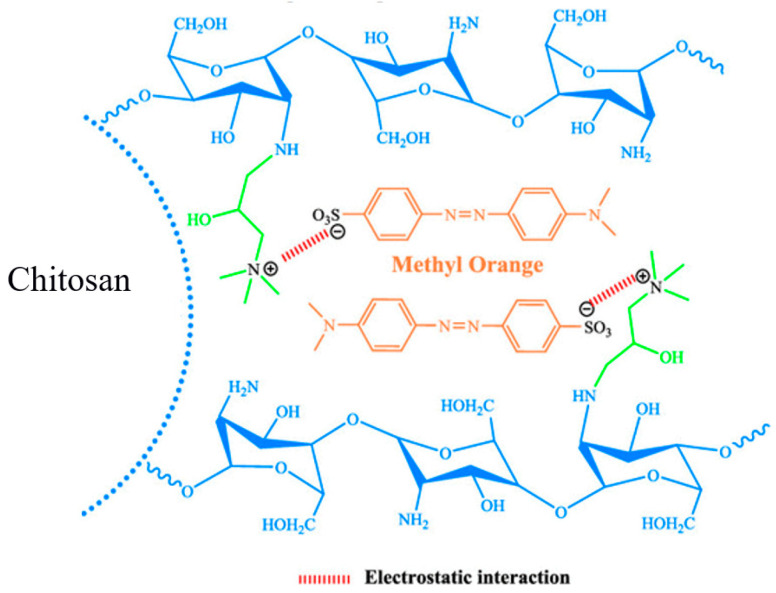
MO adsorption onto chitosan-based materials. The blue lines represent chitosan functional group. The orange lines represent MO while the green lines represent chitosan functional group involved in the interaction.

**Figure 11 molecules-28-06561-f011:**
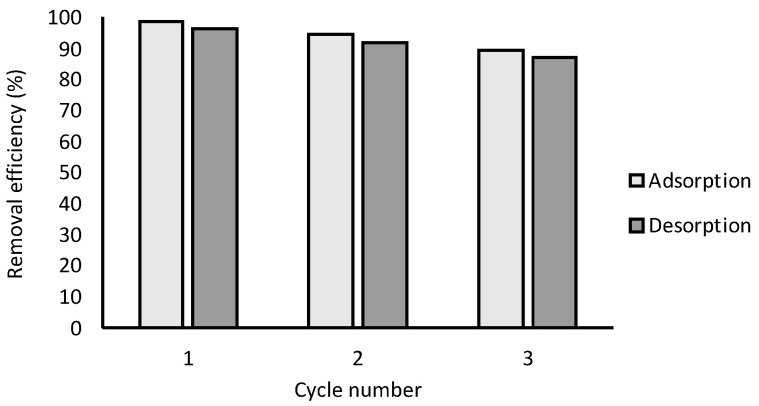
MO adsorption and desorption onto chitosan-based materials.

**Figure 12 molecules-28-06561-f012:**
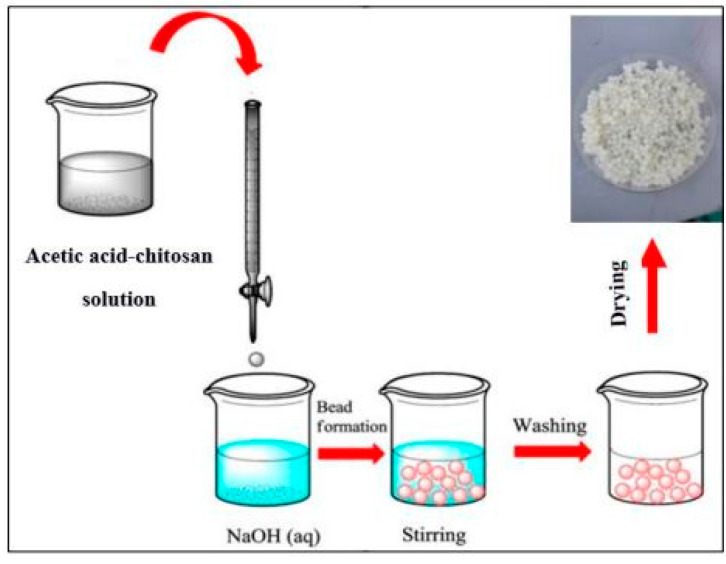
Experimental setup for chitosan bead-like material preparation.

**Table 1 molecules-28-06561-t001:** Elemental composition of the chitosan-based adsorbents.

Chitosan Adsorbents	Particle Size Diameter (µm)	Surface Aea (m^2^ g^−1^)	Total Pore Volume (cc g^−1^)
Powder	335–500	0.762	2.96 × 10^−2^
Beads	0.001	0.296	1.86 × 10^−2^

**Table 2 molecules-28-06561-t002:** Isotherm models’ constants and error analysis.

Isotherm Models	Constants	Error Analysis				
		R^2^	SSE	HYBRID	MPSD	ARE
Langmuir	q_m_ = 14.29 mg/g a_L_ = 0.371 L/mg	0.999	1.864	4.439	12.978	7.138
Freundlich	k_f_ = 5.772 L/g 1/n = 0.232	0.878	43.586	29.404	84.161	37.343
R–P	q_m_ = 18.117 mg/g k_R–P_ = 0.240 L/mg β = 1.0621	0.999	2.369	6.167	16.982	7.767
Sips	q_m_ = 20.426 mg/g k_S_ = 0.0742 L/mg 1/n = 0.0501	0.987	2.838	12.200	18.140	95.028
Temkin	B = 2.358 mg/g C = 5.005 mg/g	0.893	16.386	3.608	38.439	10.609

**Table 3 molecules-28-06561-t003:** Kinetic models’ constants and error analysis.

Kinetic Model	Constants	Error Analysis			
		SSE	HYBRID	MPSD	ARE
PFO	q_m_ = 19.602 mg/g k_1_ = 0.0928 L/mg	5.260	0.646	4.378	2.799
PSO	q_m_ = 22.130 mg/g k_2_ = 0.00588 L/mg	2.545	0.318	3.159	1.178
Elovich	β = 0.263 g/mg α = 12.577 mg/g	13.117	2.306	5.587	3.070
Intra particle diffusion	K_ID_ = 1.272 L/mg C = 9.448 mg/g	27.770	7.054	8.743	5.819

**Table 4 molecules-28-06561-t004:** Comparison of maximum adsorption capacity of various adsorbents for MO.

Adsorbents	pH	Temperature °C	q_m_ (mg/g)	References
Chitosan beads	5 7 8	-	7.2 5.8 5.6	[27]
Activated alumina	3–6	20	9.8	[37]
Banana peel	5–7	30	21	[38]
Chitosan	3	25	29	[8]
Chitosan bead-like materials	5–7	25	14.29	This study

## Data Availability

Not applicable.

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
