# Peer review of "Adsorption of Methyl Orange from Water Using Chitosan Bead-like Materials"

_molecules, 2023, doi:10.3390/molecules28186561_

Round 1

Reviewer 1 Report

The article submitted for review is devoted to the study of the adsorption of the methyl orange dye by chitosan. The topic of the article is relevant and corresponds to the profile of the journal. However, the authors need to make some clarifications.

It is necessary to provide data on the specific surface area and porosity of the studied chitosan material in Section 2.1.

The authors conclude that the maximum percentage removal of methyl orange is 64% at its initial concentration of 20 mg/l, but it is obvious that at concentrations of methyl orange tending to 0, the removal percentage will tend to 100%. This section should be supplemented with a graphical dependence of the percentage of removal on the concentration of methyl orange. It is also not clear what caused the upper limit of the studied concentration range of 100 mg/l, if the maximum solubility is about 2000 mg/l.

Missing figure 5.16 which is mentioned in the text (line 272)

Author Response

Reviewer 1:

  • It is necessary to provide data on the specific surface area and porosity of the studied chitosan material in Section 2.1.

The physical properties of chitosan raw materials and chitosan bead like materials were provided under section 3.1.1.

Chitosan adsorbents

Particle size diameter

(µm)

Surface aea

(m2 g -1)

Total pore volume (cc g-1)

Powder

335-500

0.762

2.96 E -02

Chitosan bead like materials

0.001

0.296

1.86 E -02

  • The authors conclude that the maximum percentage removal of methyl orange is 64% at its initial concentration of 20 mg/l, but it is obvious that at concentrations of methyl orange tending to 0, the removal percentage will tend to 100%. This section should be supplemented with a graphical dependence of the percentage of removal on the concentration of methyl orange. It is also not clear what caused the upper limit of the studied concentration range of 100 mg/l, if the maximum solubility is about 2000 mg/l.

This section was modified in the result and discussion section 3.2.1

The effect of various initial concentrations of MO solution on adsorption capacity are illustrated in figure 4. Increasing the MO concentration decreased the removal percentage but increased the specific adsorption capacity. The maximum removal percentage of MO dye was 98.8 %, and it was achieved with a 20 mg/L MO concentration. At low concentrations, the ratio of dye molecules to the accessible active site is low; therefore, more active sites were available for MO molecules to access. Consequently, this increases the removal percentage of MO. In contrast, fewer active sites are available for MO molecules at high MO concentrations due to the ratio of high dye molecules to the surface active sites leading to a low removal percentage but a higher dye mass adsorption capacity (Saha et al., 2010).

Although the removal percentage decreased with increasing the initial concentration, the quantitative amounts of dye adsorbed increased. With increased concentration, MO molecules electrostatically repel each other leading to a competition for the active site on chitosan. A similar effect was observed in different dye adsorption studies (Chiou et al., 2004).

  • Missing figure 5.16 which is mentioned in the text (line 272)

It was figure 8; Adsorption kinetic of MO adsorption on chitosan at variable temperature (adsorbent weight: 0.100 g, volume: 100 mL, C0= 20 mg/L).

Reviewer 2 Report

The article needed to be thoroughly revised before it get accepted for publication. Moreover the authors need to be consistent with the data presentation (please check the bar graphs). Although the data interpretation is good but need to redo the graphs and tables. The methodology needed to be revised as well. 

Reviewer 3 Report

In Fig.9, it would be better to fix the Y-axis range from 0 to 18 instead of -2 to 18.

In this manuscript, chitosan adsorption materials preparation should be discribed in detail.

Novelty should be discussed, and what are the differences between this manuscript and other chitosan-based materials?

It would be better to provide a graphical abstract for the manuscript.

Round 2

Reviewer 2 Report

I am ok the revision.